# Effects of Microorganisms in Fish Aquaculture from a Sustainable Approach: A Review

**DOI:** 10.3390/microorganisms13030485

**Published:** 2025-02-21

**Authors:** Jesús Mateo Amillano-Cisneros, María Anel Fuentes-Valencia, José Belisario Leyva-Morales, Macario Savín-Amador, Henri Márquez-Pacheco, Pedro de Jesús Bastidas-Bastidas, Lucía Leyva-Camacho, Zamaria Yoselin De la Torre-Espinosa, César Noé Badilla-Medina

**Affiliations:** 1Ingeniería en Agrobiotecnología, Universidad Politécnica del Mar y la Sierra (UPMYS), La Cruz 82700, Mexico; 2Maestría en Biotecnología Agropecuaria, Universidad Politécnica del Mar y la Sierra (UPMYS), La Cruz 82700, Mexico; 3Ingeniería en Producción Animal, Universidad Politécnica del Mar y la Sierra (UPMYS), La Cruz 82700, Mexico; 4Instituto de Ciencias Básicas e Ingeniería, Universidad Autónoma del Estado de Hidalgo, Pachuca 42184, Mexico; 5Centro de Investigación en Recursos Naturales y Sustentabilidad (CIRENYS), Universidad Bernardo O’Higgins, Avenida Viel 1497, Santiago de Chile 8370993, Chile; 6Coordinación de Ingenierías, Universidad Tecnológica de La Paz, La Paz 23088, Mexico; 7Centro de Investigación en Alimentación y Desarrollo, A.C. (CIAD), Culiacan 80396, Mexico; 8Departamento de Salud-Licenciatura en Ciencias Biomédicas, Universidad Autónoma de Occidente, Guasave 81044, Mexico

**Keywords:** probiotics, microorganisms, fish aquaculture, world, sustainable approach

## Abstract

Aquaculture is the fastest-growing food production sector. However, it faces significant challenges, including demand from a growing global population, which is estimated to reach 10.4 billion by the year 2100, disease outbreaks, environmental impacts, and the overuse of antibiotics. To address these issues, sustainable alternatives such as the use of microorganisms (probiotics, bacteriophages, and genetically modified microorganisms) have gained attention. This review examines the effects of these microorganisms on fish aquaculture, focusing on their potential to improve growth, health, and disease resistance while reducing environmental impacts. Probiotics, particularly lactic acid bacteria and yeasts, have been shown to enhance immune responses, digestive enzyme activity, and nutrient absorption in fish. Bacteriophages offer a promising alternative to antibiotics for controlling bacterial pathogens, with studies demonstrating their efficacy in reducing mortality rates in infected fish. Additionally, genetically modified microorganisms (GMMs) have been explored for their ability to produce beneficial compounds, such as enzymes and antimicrobial peptides, which can improve fish health and reduce the need for chemical treatments. Despite their potential, challenges such as regulatory hurdles, public acceptance, and environmental risks must be addressed. This review highlights the importance of further research to optimize the use of microorganisms in aquaculture and underscores their role in promoting sustainable practices. By integrating these biological tools, the aquaculture industry can move towards a more sustainable and environmentally friendly future.

## 1. Introduction

Aquaculture, defined as the cultivation of aquatic organisms (fish, mollusks, crustaceans, and algae) carried out in coastal areas and inland, is the fastest-growing food production sector. In 2022, it accounted for 50.9% of total world fisheries and aquaculture, which had a production of 185.4 million tons (MT) [1]. Of the total global aquaculture production (94.4 MT), fish represented 65.2%, followed by mollusks and crustaceans with 20.0% and 13.5%, respectively (FAO, 2024). Fish production over the years has significantly contributed to the aquaculture industry [1,2].

Fish are an important source of diverse nutrients. They have a high content of animal protein consumable by humans, they contain vitamins, essential omega-3 fatty acids, and micronutrients such as phosphorus, iron, and selenium [3,4]. Given the importance of fish consumption at the nutritional level, it is imperative to cultivate fish with better growth capacity as a way to meet the food demand of the growing global population [5], which was recorded at 8 billion and is estimated to reach 10.4 billion by the year 2100 [6].

Aquaculture activity in recent years, despite its considerable growth in terms of production, faces various challenges. Among them, sudden outbreaks of regional or imported diseases stand out, which annually cause considerable economic losses estimated at USD 6 billion on a global scale [7,8]. These disease outbreaks are caused by viruses (white spot syndrome virus, yellowhead virus, infectious salmon anemia virus, salmonid alphavirus, Tilapia lake virus, iridoviral disease, kidney necrosis virus), bacteria (*Aeromonas* spp., *Edwardsiella* spp., *Flavobacterium* spp., *Streptococcus* spp., *Vibrio* spp.) fungi (*Achlya* spp., *Aphanomyces* spp., *Saprolegnia* spp., *Batrachochytrium* spp., *Branchiomyces* spp., *Ichthyophonus* spp., *Fusarium* spp.), and parasitic pathogens from different taxa (*Amyloodinium* spp., *Ichthyobodo* spp., *Ichthyophthirius* spp., *Myxobolus* spp., *Tetracapsuloides* spp., *Tetrahymena* spp., *Trichodina* spp., *Trypanoplasma* spp., *Trypanosoma* spp., *Uronema* spp.) [7,9,10,11,12,13,14,15,16,17,18].

Another challenge currently reported for fish farming in marine and freshwater environments is the presence of co-infection [19,20,21]. Co-infection refers to an infection caused by two or more pathogens, including infectious agents of different taxonomic and genetic variants [22]. Co-infection with pathogenic microorganisms has negative effects on susceptible fish, such as increased mortality and disease severity. In many cases, one pathogen is considered critical and therapy is recommended, while co-infecting agents are ignored, hindering the effectiveness of treatment and compromising the health of aquatic organisms [22,23].

Frequently, with the presence of these sudden outbreaks and the lack of experience in disease management, they cause significant damage to production in terms of low growth rates, high mortality, and even environmental degradation, as they are vectors of diseases in the aquatic environment that can spread to farms in the vicinity where the disease is present or even to natural environments where such diseases are not found also causing damage to native aquatic animals [8,12,13,24,25].

Also, due to the increase in demand for aquatic products because of their nutritional quality, the use of intensive-type farming is encouraged year after year, which has the characteristic of producing large quantities of food in small spaces. However, they are considered to have negative impacts on the environment, such as the release of high levels of nitrogen and phosphorus derived from their metabolism, which causes eutrophication of the water around the cultivation area [5,26,27,28]. Another significant environmental impact is that when an intensive production area is chosen, systems with high ecological function, such as mangrove forests, are destroyed [29,30,31].

Within intensive aquaculture production systems, overcrowded organisms suffer from chronic stress conditions, which negatively affect their health, making them susceptible to acquiring opportunistic viral, fungal, bacterial, and parasitic diseases that can even be transmitted to healthy fish and cause considerable production and monetary losses [32,33,34]. To avoid these overcrowding-related diseases, aquaculture producers frequently use a considerable variety of antibiotics when there is illness [35,36] and even apply them preventively to healthy organisms [14,37,38]. There is a list of antibiotics that are approved for aquaculture production [39]. However, the indiscriminate use of these products in aquatic environments leads to the spread of their residues, which can result in increased resistance in aquatic bacteria with pathogenic potential and can affect the health of human consumers through disease transmission [26,37,40,41].

Due to the various challenges faced by this productive activity, alternatives related to the concept of sustainable development must be investigated so that its growth remains steady and contributes to the global food supply. The Brundtland Commission conceptualizes sustainable development as follows: “sustainable development is development that meets the needs of the present without compromising the ability of future generations to meet their own needs” [42,43].

A viable sustainable alternative for aquaculture is the use and application of microorganisms in fish diets, with the aim of defining and establishing their multiple beneficial effects in terms of improving the production and health of these organisms. Kenis et al. [44] defines microorganism as “a protozoan, fungus, bacterium, virus, or other microscopic self-replicating biotic entity”. However, Shams et al. [45] mention that microorganisms themselves are not clearly defined scientifically, although they define them as microscopic-sized organisms with rapid generation times and as including species from all three domains of life (Archaea, Bacteria, and Eukarya). Since their discovery in the 17th century by Anton Van Leeuwenhoek, the work regarding the discovery of new species, characteristics, and their functions has been constant to this day.

Microbiota is defined as the collective community of microorganisms (viruses, bacteria, archaea, yeasts, and protozoa) that inhabit a specific ecosystem or environment (e.g., skin, gastrointestinal tract, water, soil) [46,47]. In organisms such as fish, it has been established that gut microbiota have various important functions, such as protection against pathogens, enhancement of the immune system, as well as the synthesis of metabolites (vitamins, minerals, production of short-chain fatty acids and amino acids) that can be available to be acquired as nutrients by the host, to grow and develop properly [48,49,50,51].

There is a large number of studies that address the composition of microorganisms as part of the intestinal microbiota and the different effects they have on aquatic organisms such as fish, which represent a nutritional, economic, and viable source in terms of production quantity since they contribute a considerable percentage (>65%) to the total global aquaculture production. However, most of this type of work presents the importance of the use of bacteria, yeasts, and viruses separately, without considering the importance that these types of microorganisms together represent an important alternative for the continuous development of this activity, which tends to present continuous problems of disease outbreaks that cause considerable production and economic losses, as well as resistance to substances such as antibiotics. The present work aims to emphasize the need to develop a better integrative understanding of the beneficial effects of applying viruses, bacteria, and yeasts in fish diets, originating from various sources such as terrestrial (fermented) foods, soil, sediment of aquatic environments, the water where these organisms are found, other species or taxa, and even the same species of fish, representing a sustainable alternative in improving cultivation conditions. This document defines the updated studies on the beneficial effects of applying bacteria, yeasts, and viruses to the diet of the most commonly used fish in global aquaculture and analyzes the current use of bacteriophages in fish farming as a potential sustainable alternative in the presence of bacterial diseases that cause severe annual losses worldwide. Additionally, we present the option of using genetically modified microorganisms in fish aquaculture as another sustainable alternative in a growing aquaculture. It is necessary to mention that having a comprehensive understanding of the use of microorganisms, both isolated from different media and genetically modified in aquaculture, will allow us to have a clearer idea of where future studies in this area of food production could be directed.

## 2. Methodology Applied for the Literature Review

The literature search strategy in this research was based on the PRISMA model (Preferred Reporting Items for Systematic Reviews and Meta-Analyses). This methodology requires reporting each step of the article selection process (identification, inclusion, and exclusion). It includes constructing a flow diagram that documents the number of studies evaluated at each stage [52,53,54].

The investigation is guided by the following key question: “What effects of microorganisms in fish aquaculture from a sustainable approach according to the scientific literature?” This question ensures that the review is specific and addresses a well-defined problem. The keywords were also defined: EM (effects of microorganisms), FA (fish aquaculture), P (probiotics), B (bacteria effects), Y (yeast effects), V (bacteriophages effects), GMM (genetically modified microorganisms) and SA (sustainable approach). The search was performed in databases such as Scopus and ScienceDirect, and different combinations of the keyword strings were used, covering a period of 23 years (2002–2024) with some exceptions, obtained using the snowball approach, where older references from books or articles were used to cite classic definitions despite not fulfilling some of the criteria explained above (Figure 1). In summary, the inclusion and exclusion criteria were as follows. In the first criterion, the quality of the document consulted in the databases was considered. If it was not of high quality, it was rejected. On the other hand, if it was of high quality, it was passed on to the next criterion. The second criterion consisted of analyzing consecutively the title of the document, the summary and the whole document. Each of these points was analyzed in terms of its potential relevance to the sections written in the review article. If any of these consecutive points were not relevant, the document was rejected. Conversely, if it was relevant, it was moved on to the next and final criterion. The third criterion was the complete analysis of the document, from which relevant aspects for the development of the different sections of the article were obtained.

The results obtained for the different keyword combinations are shown in Table 1, which includes the number of papers related to the keyword search and present in the Science Direct and Scopus databases, the sections where the keyword searches were applied, as well as the number of documents included in the different sections of this review. The search was limited to books and both review and research articles were considered. Only manuscripts in English were included.

## 3. Effects of Microorganisms on Fish Aquaculture

Microorganisms from various species of bacteria, yeasts, and families of viruses have been applied over the years as probiotics in fish aquaculture. Probiotics have been generally defined as live microorganisms that, in adequate doses (1 × 10^6^–1 × 10^7^ CFU/g), have beneficial effects on organisms that consume them [55]. However, in aquaculture, there is a variation in this concept where the differentiation between terrestrial and aquatic environments is taken into account, defining probiotics as “an organism that can be considered alive, dead, or a component of a microbial cell, which administered via feed or rearing water, benefiting the host by improving disease resistance, health status, growth performance, feed utilization, stress response, or general vigor, which is achieved at least in part by improving the microbial balance of the host or the microbial balance of the environmental setting” [56].

Among the most common sources of probiotics used in aquaculture are various types of microorganisms (bacteria, microalgae, viruses, molds, and yeasts) and macroorganisms like fungi and plants [41,57,58].

It has been proven that the application of viruses, bacteria, and yeasts in the diet of fish used in aquaculture has multiple beneficial effects, such as increased growth, survival, and immune system, various nutrients contained in fish muscle, modulation of microbiota and intestinal morphology, as well as protection against diseases caused by pathogenic microorganisms [59,60,61] (Figure 2).

### 3.1. Effects of Bacteria in Fish Aquaculture

Among the probiotics used in fish aquaculture, bacterial probiotics include species of the genera *Lactobacillus*, *Lactococcus*, *Streptococcus*, *Bacillus*, *Enterococcus*, *Alteromonas*, *Arthrobacter*, *Bifidobacterium*, *Clostridium*, *Paenibacillus*, *Phaeobacter*, *Pseudoalteromonas*, *Pseudomonas*, *Shewanella*, *Rhodosporidium*, *Roseobacter*, and *Streptomyces* [41,62,63,64,65]. Furthermore, some potentially pathogenic bacterial strains from the genera *Aeromonas* and *Vibrio* have been successfully applied as probiotics [66,67].

In the past twenty years, most of the results from studies on the use of bacteria in fish of global production and economic importance have primarily focused on growth improvement, which is one of the key points for increasing and maintaining global levels of farmed fish production. Bacteria mainly from the genera *Bacillus*, *Lactobacillus*, *Lactococcus*, *Shewanella*, and *Streptococcus* have shown to primarily improve the growth of fish species, such as *Acipenser baerii*, *Carassius auratus*, *Ctenopharyngodon idella*, *Cyprinus carpio*, *Oncorhynchus mykiss*, *Oreochromis niloticus*, *Pangasius hypophthalmus*, *Paralichthys olivaceus*, and *Solea senegalensis*. However, these studies have also shown effects on increasing survival, production of digestive enzymes (protease, amylase, and lipase), and enzymes related to the immune system (e.g., lysozyme), improving intestinal health and regulating intestinal microbial by increasing lactic acid bacteria and reducing potentially pathogenic bacteria (Table 2).

There is currently a large number of investigations on the application of lactic acid bacteria on the improvement of survival, growth, disease resistance, and feed efficiency. For example, in *Dactylopterus volitans* larvae, *Lactococcus lactis* PH3-05 at a concentration of 1 × 10^4^, 1 × 10^6^ and 1 × 10^8^ CFU/g is isolated from the intestine of an adult tropical fish. It should be noted that the dose of 1 × 10^8^ CFU/g stimulated a greater expression of the muc-2 and il-10 genes, suggesting an improved mucosal barrier function and an anti-inflammatory response. The dose of 1 × 10^6^ CFU/g significantly improved survival by 46% and the activity of digestive enzymes, so we can say that it has been shown to significantly improve growth, survival, and digestive enzyme activity [79]. Similarly, in *Oreochromis niloticus*, administration of *Lactococcus lactis* at a concentration of 1 × 10^8^ CFU/mL to sole improved non-specific immune parameters such as lysozyme, antiprotease, serum peroxidase and blood respiratory burst activities. Nine days after the challenge with *Streptococcus iniae* (1 × 10^8^ CFU/mL), the untreated control group experienced a 90% mortality rate, while all fish supplemented with *L. lactis* survived. Thus, *L. lactis* has shown positive results by increasing weight gain and survival rate [80]. The use of probiotics has also been highlighted in improving innate immunity. In *O. niloticus*, combinations of *Bacillus subtilis* and *Lactobacillus casei,* administered at concentrations of 1 × 10^8^ CFU/mL, contribute to the production of protective antigens; that is, the strains used at 15% increased phagocytic activity in the group of infected fish (*Aeromona hydrophila)* after infection [81] (Table 2).

In *Pangasius bocourti*, *Bacillus aerius* B81e and *Lactiplantibacillus paraplantarum* L34b-2 at a concentration of 1 × 10^7^ CFU/g were used. The probiotic *B. aerius* improved fish growth probably due to its ability to produce protease and lipase enzymes and *L. paraplantarum* L34b-2′s ability to produce protease [82]. The significant reduction in feed conversion ratio (FCR) demonstrated that fish more efficiently utilized dietary nutrients when supplemented with mixed probiotics. In contrast, Xia et al. [97] reported that *O. niloticus* fed a probiotic mixture composed of *Lactococcus lactis* subsp. *Lactis* JCM5805 and *Lacticaseibacillus rhamnosus* JCM1136 did not show superior growth compared to the control group. However, Hooshyar et al. [88] demonstrated how the design and development of a diet containing encapsulated probiotics (*Lactobacillus rhamnosus* ATCC 7469) more effectively improved the survival rate of rainbow trout (*Oncorhynchus mykiss*) after the *Yersinia ruckeri* challenge test. In addition, diets supplemented only with encapsulated *L. rhamnosus* ATCC 7469 positively influenced growth performance, body composition, blood biochemistry, antioxidant activity, and the immune system of rainbow trout. In *Rachycentron canadum*, supplementation with *B. cereus* increases growth and activity of digestive enzymes [83]. In this same sense, Adeshina et al. [86] showed that the *Lactobacillus acidophilus* strain administered to the common carp *Cyprinus carpio* at a concentration of 1 × 10^9^ CFU/g had weight gain, a stable growth rate (SGR), feed intake, and a feed conversion ratio (FCR) significantly higher than those in the control group; however, the innate immune profiles, superoxide dismutase, catalase, respiratory enzyme activity, as well as transforming growth factor beta (TGFβ), interleukin 8 (IL-8), and tumor necrosis factor alpha (TNF-alpha) were also significantly stimulated. On the other hand, Akbari et al. [87] showed that oral administration of *Enterococcus casseliflavus* (EC-001) had a beneficial effect in improving growth and non-specific immune responses of common carp fingerlings. The best growth performance and feed conversion ratio were observed in fish fed with *E. casselifavus* (EC-001) at 1 × 10^9^ CFU/g. In addition, an improvement in hematological parameters and humoral and skin mucosal immune responses was obtained in the treated fish compared to the control group (Table 2).

Silva et al. [89] used a probiotic mixture (*Bacillus, Bifidobacterium, Enterococcus, Lactobacillus, Pediococcus*, *B. subtilis*) at a concentration of 7 × 10^10^ cells/kg as a feed additive. It showed that the performance of *O. niloticus* improved, as well as the feed conversion rate, the final weight, and the growth. They concluded that the stimulation of performance and development occurred due to an increase in the number of intestinal villi, guaranteeing a greater absorption and utilization of the nutrients provided by the diet, which was reflected in the hypertrophy of the white muscle fibers accompanied by the inhibition of the expression of the MyoD (Myogenic Differentiation) and myostatin genes (Table 2).

The nutrients contained in fish and other aquaculture species (proteins, carbohydrates, fatty acids (omega-3), vitamins, minerals, among others) are important for the food industry because they are transmitted to humans and are essential for human health due to their protective antiarrhythmic, antithrombotic, antisclerotic, anti-inflammatory, antidiabetic, anticancer, and antioxidants, among other properties [98,99]. Furthermore, an adequate amount of nutrients in the aquaculture species diet is also important for their nutrition in culture conditions, as they help them grow, have good health, and be resistant to exposure to pathogens [90,100]. In addition to the main nutrients, the aquaculture species diet requires the introduction of biologically active substances produced by microorganisms (probiotics) that favor intensive fish production.

Although there have been numerous studies worldwide where probiotics have been applied to the commercial fish species, it is appropriate to mention some of those carried out. For example, when including the probiotic Sporothermine (spore forms of the bacteria *Bacillus subtilis* and *Bacillus licheniformis*) in the fish diet of African catfish *Clarias gariepinu*, an increase in the content of some vitamins was observed [90] (Table 2). In another study, using *Lactococcus lactis* K-C2 as probiotic in the feed of Amberjack *Seriola dumerili,* the amount of 13 amino acids in the edible parts of fish were significantly higher [61] (Table 2).

In the Nile tilapia *Oreochromis niloticus*, Hamdan et al. [73] applied the probiotic *Lactobacillus plantarum* AH 78 isolated from corals samples along the Egyptian coasts in a concentration of 1% feed, founding a significant increase in total protein of muscle fish (Table 2). Other studies have reported that the increase in short chain fatty acids (SCFA) derived from the application of probiotics in diet of cultured fish, such as Javanese carp *Puntius gonionotus* [91] and Caspian Roach *Rutilus frisii kutum* [80]. This has also been observed in long-chain fatty acids (LCFA), particularly Docosahexaenoic acid (DHA) and Eicosapentaenoic acid (EPA), due to the addition of probiotic agents in fish diet [93,94] (Table 2).

The application of probiotics in aquaculture favors the improvement of survival, growth, disease resistance, feed efficiency, and presence of macro- and micro-nutrients. Also, within the host, intermediate products or metabolites are produced that help the organism in various activities, such as defense against pathogens. Among the most reported products or metabolites are bacteriocins, amino acids, volatile fatty acids, and antimicrobial peptides [95,96,97] (Table 2). However, yeasts also produce substances or metabolites capable of preventing the proliferation of bacteria and other pathogenic organisms.

### 3.2. Effects of Yeasts in Fish Aquaculture

Yeasts are unicellular eukaryotic microorganisms and are part of the normal microbiota of fish [63]. Yeasts have been widely used in probiotic assessments for fish aquaculture and include species from genera *Debaryomyces, Rhodotorula, Saccharomyces,* and *Yarrowia* [41,62,101,102,103,104] (Table 3).

The use of probiotics based on yeasts is gaining ground in aquaculture due to its multiple benefits for the health and performance of species such as *O. niloticus*, *Cyprinus carpio,* and *Sparus aurata*. The widely used probiotic *Saccharomyces cerevisiae* in *O. niloticus* has shown significant benefits in different contexts. Abd et al. [110] reported an increase in final body weight and weight gain when fed between 1 and 3 g/kg for eight weeks, with diets supplemented with Hilyses^®^ at concentrations of 0, 1, 2 or 3 g/kg, this is because dietary Hilyses decreased amylase activity, but increased protease and lipase to varying degrees. Improvements in protease and lipase activity may improve protein and lipid digestion, allowing for greater nutrient absorption and promoting growth. While Akanmu et al. [103] observed that the inclusion of 3% of the probiotic yeast *S. cerevisiae* obtained from palm wine, improved growth and blood components, and when the fish was challenged against pathogen *Aeromona hydrophila*, the yeast treatment improved the resistance. Additionally, the combination of yeast *Saccharomyces boulardii* (1 × 10^10^ CFU/g) and bacteria *Bifidobacterium bifidum* (1.5 × 10^8^ CFU/mL) has been shown to improve immune responses and alleviate heat stress and oxidative damage of *O niloticus* [111]. Also, *S. cerevisiae* has been applied with positive results in Cyprinids *Carassius auratus gibelio* and *Cyprinus carpio,* improving survival in the presence of pathogen *Aeromonas hydrophila* and growth performance parameters, respectively [105,106] (Table 3).

Other probiotics yeasts applied in fish farming have been *Debaryomyces hansenii, Yarrowia lipolytica* and *Rhodotorula mucilaginosa*. Reyes-Becerril et al. [112] found that the application of *D. hansenii* leopard grouper *Mycteroperca rosacea* improves the immune system and resistance against the pathogen dinoflagellate *Amyloodinium ocellatum*. Menawhile Sanahuja et al. [102] found that by applying the same yeast to gilthead seabream, *Sparus aurata* improves growth and reduces the abundance of opportunistic bacteria *Pseudomonas* spp. and *Acinetobacter* spp. *Yarrowia lipolytica* has been tested as a probiotic in marine and freshwater fish. In Pacific red snapper *Lutjanus peru,* it improves the innate immune and antioxidant enzyme activities in the presence of the pathogen *Vibrio parahaemolyticus* [114]. For freshwater fish, such as Nile tilapia and rainbow trout, the yeast improves the protein contents in the muscles and the expression of immune genes, respectively [104,115]. The yeast *Rhodotorula mucilaginosa* has been proven to improve growth and resistance survival rate against pathogens in *O. niloticus* and *Trachinotus ovatus* [101,117]. Also, innovative probiotics, such as *Sporotrichoides petaronensis,* have been evaluated with promising results. In *O. niloticus*, *S. petaronensis* improved growth parameters and immune response against the pathogen *Streptococcus agalactiae* when administered at doses of 1% and 2%, for up to 90 days [118] (Table 3).

Yeasts have shown significant potential on improving fish health and performance through their ability to modulate the immune system and enhance nutrient absorption. However, the use of microorganisms in aquaculture is not limited to bacteria and yeasts. Another promising alternative is the application of bacteriophages, which offer a sustainable solution for controlling bacterial pathogens without the risks associated with antibiotics.

### 3.3. Effects of Virus in Fish Aquaculture

Within the group of viruses, bacterial viruses or bacteriophages (commonly phages) are the most abundant biological entities on Earth [120], and they are present in both freshwater and marine environments [121,122]. They were independently discovered before antibiotics by Frederick Twort in 1915 and Félix d’Hérelle in 1917 [123]. There is a global shortage of innovative antibiotics that are effective against pathogens that are resistant to them. The World Health Organization has recently pointed out that none of the 97 antibiotics in clinical development sufficiently address the problem of drug resistance in the world’s most dangerous bacteria [124], so there is renewed interest in alternatives such as bacterial viruses.

Bacteriophages are viable biocontrol agents that can be applied as both prophylactic and therapeutic measures against bacterial infections [125]. In addition, they are obligate intracellular parasites of bacteria, and, based on their replicative cycles, they are classified as lytic and lysogenic [123]. Furthermore, bacteriophages are abundant in aquatic environments and persist for long periods [126]. The search for alternative strategies has considered bacteriophages as potential therapies in recent years to reduce the emergence of antimicrobial-resistant (AMR) bacterial strains [127]. Phage therapy is the use of bacteriophage viruses (phages), which infect and lyse specific bacteria for the control of infectious diseases [128].

The indiscriminate use of antibiotics globally in animal production is one of the main causes of the rapid spread of antimicrobial resistance [129], which are used as growth promoters or as prophylactics to increase animal productivity [130,131]. In particular, lytic bacteriophages have the potential for specific control of pathogenic bacteria without a negative impact on the environment compared to antibiotics; they are, therefore, recognized as important biotherapeutic agents [132]. Several studies demonstrate the usefulness of lytic bacteriophages in control of different types of pathogenic bacteria in fish and marine organisms [133,134,135,136]. Bacteriophage-derived endolysin therapy employs hydrolytic endolysin enzymes that target bacterial peptidoglycan cell walls. It was recently reported as an innovative method for disease control in aquaculture for the control of *Streptococcus iniae,* achieving a 95% survival rate in hybrid striped bass [137].

The success of phage therapy in different studies opens a new window to apply its concepts in aquaculture, especially in fish and crustaceans [138]. In particular, there are reports on the application of phage therapy for various Gram-negative and Gram-positive pathogens affecting fish farms with different modes of application evaluated (oral feeding, immersion, bath and intraperitoneal injection) (Table 4). Some investigations simultaneously compared these routes of application of phage therapy in fish such as Muliya et al. The authors of [136] evaluated the phage AhFM11 of the *Straboviridae* family against *A. hydrophila in Labeo rohita,* using three administration methods: injection, immersion, and oral feeding, with a specific concentration for each method. Administration via injection resulted in 100% survival, while immersion and oral feeding achieved rates of 95% and 93%, respectively, evidencing the versatility of the phage in different application contexts. In another in vivo experiment on rainbow trout, intraperitoneal injection treatment of bacteriophages, an 80% survival rate was recorded compared to the control group (57%). It is, therefore, suggested that higher doses of phages via feed are required to protect against bacterial infection [139]. However, in other recent research, bath treatment was the most protective against bacterial infection, with an 80% survival rate compared to intraperitoneal and oral applications with 70% and 50%, respectively [125]. Intraperitoneal application is the main mode of application evaluated in experimental research, particularly with 100% protective effects in fish against *Aeromonas hydrophila* [136,140].

Currently, phage therapy has gained biotechnological interest as a biocontrol strategy compared to antibiotics and vaccines [125]. In different studies, bacteriophages are reported to present several advantages, such as bacterial specificity, self-limiting capacity, self-dosing, and adaptation to resistant bacterial species [129,149,150,151,152]. Therefore, phage therapy represents a sustainable opportunity to protect aquatic animal health, which presents several challenges, given that antibiotics are the primary treatment of choice and present several conditions for their ineffectiveness in aquaculture, such as inadequate concentrations in target organs, innate or acquired bacterial resistance to antibiotics, inadequate therapeutic durations, presence of bacterial co-infections, misdiagnosis [153], as well as a lack of adaptation to resistant bacterial species [151]. Furthermore, the antibiotics permitted by the U.S. Food and Drug Administration (FDA) [39] are in short supply with four of them. With the challenge of increasing animal protein production on a global scale and where aquaculture is an important contributor, alternatives are needed, and bacteriophages could be an option for sustainable use, as they are an inexhaustible resource for the treatment of bacterial infections in this sector and they do not present the ravages of antibiotics.

There is a need for further experimental research in fish farming to continue the exploration of various routes of application, as the results reported in numerous investigations are promising. However, there are practical implications, e.g., to identify the efficient and optimal mode of application, as intraperitoneal injection is one of the effective routes of application; on the other hand, in larger scale installations, it would be impractical for animal management.

With time and accumulated research in this topic, it is expected that in the medium term, the treatment of bacterial diseases using bacteriophages will become more feasible and practical in terms of breadth of effect on different species of primary and secondary bacterial pathogens as well as therapeutic cost.

Bacteriophages have emerged as a promising tool for controlling bacterial infections in aquaculture, offering a targeted and environmentally friendly alternative to antibiotics. However, the potential of microorganisms in aquaculture extends beyond probiotics and bacteriophages. Genetically modified microorganisms (GMMs) represent a cutting-edge approach to enhancing fish health and production through the expression of beneficial genes and proteins.

## 4. Use of Genetically Modified Microorganisms in Fish Aquaculture

In addition to beneficial effects of isolated microorganisms of different sources applied to the diets of fish, there are genetically modified microorganisms (GMMs), which are categorized as part of genetically modified organisms (GMOs). Both GMOs and GMMs can be considered to have sustainable potential in terms of their use in the aquaculture industry. By definition, GMOs are organisms that have undergone scientific alteration of their genetic material, which include microorganisms (e.g., bacteria, yeasts), insects, plants, fish, and mammals [5]. Within GMOs, the mouse was the first transgenic animal [5,154]. In fish aquaculture, the first genetic editing works were performed on rainbow trout [155], catfish [156], and tilapia [157]. Currently, fish aquaculture and experimental importance from other species, such as zebrafish, Atlantic and coho salmon, tilapia, common carp, catfish, red sea bream, and tiger pufferfish, are produced with the aim of increasing production efficiency and being able to use it commercially [5,158,159,160].

Within GMOs, genetically modified microorganisms (GMMs) have applications in human health through vaccines [161], therapeutic proteins [162], use in therapy for chronic diseases [163], as well as their use for decades as bioreactors to generate molecules like insulin for diabetes treatment [164]. In agriculture, GMMs are used to stimulate plant growth, increase nutrient availability, as well as for the protection and treatment of plant diseases and pests [163,165,166]. While in ecology, they are used for wastewater treatment as bioremediation [167,168,169].

Regarding fish aquaculture, bacteria, yeasts, and viruses, both probiotic and potentially pathogenic, have been collected and analyzed with the aim of cultivating them, molecularly identifying them to edit genes that generate proteins and resistance factors (epitopes), which, once applied by injecting it into food or the water where the fish are located, mainly have effects on improving survival and the immune system in the presence of pathogenic bacteria and viruses (Figure 3).

There are studies on the use of genetically modified bacteria, in which they analyze their potential as a sustainable alternative in the aquaculture food industry. In crustaceans, it has been demonstrated that genetically modified bacteria of the species *Escherichia coli*, *Bacillus subtilis*, and *Lactobacillus plantarum* have been used as bioreactors to generate antiviral molecules for the treatment of viral infections in shrimp [170,171,172]. Additionally, a native strain of *B. cereus* isolated from *Ucides* sp. crab has been genetically manipulated and applied to white shrimp *Litopenaeus vannamei* to determine its effect on weight and survival without statistical differences between the control treatments and the bacteria [173]. However, this type of study demonstrates the potential use of genetically edited native probiotic strains to be applied in aquaculture.

Regarding fish aquaculture, the use of genetically modified microorganisms is in its early stages of development and focuses on modifying both probiotic and pathogenic microorganisms.

Lactic acid bacteria probiotics have been genetically edited to produce the enzyme phytase, which is experimentally added to fish diets to improve phosphorus digestibility, subsequently improving its assimilation for better animal growth while reducing phosphorus excretion and, consequently, minimizing environmental pollution as an alternative to sustainable animal production [174]. In another study, a strain of the potential probiotic *B. subtilis* isolated from the intestine of Nile tilapia was genetically edited and applied to the same fish species to express CC-Chemokine with the aim to stimulate the immune response [175]. Another possibility in this area of production is the genetic editing of pathogenic bacteria, such as *Flavobacterium psychrophilum*, which causes significant damage in salmonid fish [176]. The editing of this microorganism is considered a challenge, as the editing techniques to generate attenuated vaccines from the different strains of this pathogen are particularly complex, although it can be considered as a guide for use in aggressive pathogenic microorganisms in aquaculture [177]. Another genetically modified pathogenic bacterium has been *Aeromonas hydrophila*, where Poobalane et al. [178] applied this bacterium with a recombinant protein to evaluate the protective capacity in common carp *Cyprinus carpio* against six virulent isolates of *A. hydrophila*, resulting in survival with significant differences compared to the control fish, thus demonstrating its protective capacity against pathogenic bacterial strains.

Although the studies are scarce, lactic acid bacteria have recently also been used as recombinant probiotics [41], which could represent another sustainable alternative for application in aquaculture for the treatment of both viral diseases and those caused by highly pathogenic bacteria. In summary, the method of using this alternative is to attack these diseases by applying an epitope or antigenic determinant to probiotics, which, after being applied to the diet of commercially important fish, the immune system of these organisms recognizes and adapts to when the harmful virus or bacteria attacks. For example, in rainbow trout *Oncorhynchus mykiss*, the application of *Lactobacillus casei* expressing different epitopes of infectious pancreatic necrosis virus (IPNV) provides subsequent protection against the same virus [63,179,180]. Similarly, *L. casei* expresses epitopes to develop immunization against spring viremia carp virus (SVCV) in common carp *Cyprinus carpio* [181]. Moreover, in the carp species *C. carpio* and *Carassius carassius*, the same recombinant bacteria express epitopes of pathogenic bacteria *Aeromonas veronii*, *A. hydrophila*, and *Vibrio mimmicus*, resulting in beneficial effects on the increase in survival with the presence of these bacteria [182,183,184,185,186]. Another recombinant bacterium that has been used in carp is *L. plantarum*, which expresses epitopes against SVCV, koi herpesvirus (KHV), and *A. hydrophila* [187,188,189].

*Escherichia coli* has also been used as a recombinant microorganism. Aonullah et al. [190] immersed juvenile *Cyprinus carpio* in water containing heat-killed *Escherichia coli* carrying the gene for glycoprotein-25 present in Koi herpesvirus (KHV); this resulted in a higher survival percentage of fish challenged by KHV as well as the detection of anti-KHV antibodies.

Regarding genetically edited microscopic yeasts, the potential of genetic editing on the yeast *Saccharomyces cerevisiae* has been analyzed. Luo et al. [191] expressed the VP7 protein of the grass carp reovirus (GCRV) in this yeast with the aim of developing a functional vaccine against hemorrhagic disease in herbivorous carp. Zhao et al. [192] designed an oral vaccine from this genetically edited yeast species to express the glycoprotein (G) of the Infectious Hematopoietic Necrosis Virus (IHNV) and successfully administered it orally to rainbow trout, where it expressed immune response markers, and the survival rate upon IHNV exposure was around 50%. Also, *S. cerevisiae* has been studied with the aim to develop strains that have higher protein production as an alternative to conventional protein sources for feeding farmed fish [63,193]. Additionally, other yeast species have been studied for their potential in genetic editing. The yeast *Pichia pastoris* has been edited to generate proteins from the capsid of reds-potted grouper nervous necrosis virus (RGNNV) with the aim to develop an oral vaccine against the same virus when it infects fish [194]. While the oleaginous yeast *Yarrowia lipolytica* was used in fish and crustacean diets to improve health and the immune system [104,195,196], by inactivating the peroxisome biogenesis gene (PEX10), it enabled high eicosapentaenoic acid (EPA) yields [197]; this can enhance its beneficial effects when applied in aquaculture industry.

Regarding the use of genetically edited viruses and bacteriophages for use in fish aquaculture, the potential has been mainly analyzed for viruses that significantly damage the main aquaculture species, such as fish of family Salmonidae and Sinipercidae families. The genetic editing of pathogenic viruses through targeted mutations is an interesting way to apply attenuated viruses and develop effective vaccines [198], because an attenuated virus, when correctly applied, can stimulate the immune system and protect the organism when the pathogen is present.

In fish, vaccines are used to stimulate the immune system and provide protection against specific pathogens. Immersion and injection are the two main methods of administering vaccines to fish [199]. Highly pathogenic virus vaccines, such as the viral hemorrhagic septicemia virus (VHSV), which are one of the main causes of mortality in various species of freshwater and marine fish worldwide, have been studied and developed. However, research on mutated vaccines of this virus and their effects on improving immunity in fish and their use in aquaculture farms is still lacking [200]. In contrast, Moriette et al. [201] infected juvenile rainbow trout with recombinant and wild-type sleeping disease virus (SDV), a member of the Salmonid alphavirus genus within the *Togaviridae* family. Among their results, when trouts were infected by immersion in a bath of water with recombinant SDV, there was no mortality or signs of disease in the fish. However, the organisms infected with the wild-type SDV (wtSDV) reached an accumulated mortality of 80%. Moreover, 3 and 5 months after the rSDV infection, the fish were challenged with the wtSDV and showed lasting protection against this wild virus. In another experiment applied to salmonids, Aksnes et al. [198] analyzed the effect of immunizing Atlantic salmon with three attenuated infectious strains with targeted mutations of the salmonid alphavirus (SAV), which causes serious problems in European salmonid aquaculture. Among its results, it was highlighted that in fish immunized through injection, the strain of SAV used for the challenge against the virus was not detected, indicating that the fish showed immune adaptation against superinfection with SAV during the 12 weeks of the experiment. Zeng et al. [202] noted that the attenuated gene ΔORF022L gene of infectious spleen and kidney necrosis virus (ISKNV), which causes high mortality and economic losses in aquaculture in Asia, was studied as a potential vaccine for mandarin fish *Siniperca chuatsi* to protect it against the same virus. As a result, a 100% survival rate was observed in the fish pre-infected with ΔORF022L and then infected with ISKNV. Additionally, ΔORF022L in fish increased immunity-related genes and generated specific antibodies against ISKNV.

Immersion and injection are vaccines commonly administered to fish. Immersion is often available for a small number of bacterial pathogens and is difficult to use on large fish [199,203]. As the injection requires the fish to be of a certain size, it makes it practically impossible to vaccinate fry or large quantities of fish [199]. Other disadvantages of vaccination are the stress caused to the organisms by handling, the processing time, and the monetary cost. In addition to the fact that vaccination immunity can only protect fish in the short term [199], it is not cost-effective to develop vaccines against all known pathogens when they are identified. Therefore, fish vaccination in aquaculture focuses on pathogens that cause significant mortality [204]. Moreover, in Mondal and Thomas’s work [204] on recent advances and the application of vaccines against fish pathogens in aquaculture, they indicate that vaccines produced in the future to treat bacterial and viral pathogens in fish must be environmentally friendly and cost-effective for large-scale production so that they are available to all types of fish producers, including small-scale ones.

Despite recent studies on experiments involving the application of genetically modified microorganisms in aquaculture, they are still scarce. In addition, most of the experimental work on GMMs applied in fish aquaculture is focused in genetically editing pathogenic microorganisms, as they are the ones that cause the main production and monetary problems, and it is often difficult to control or eliminate them with common antibiotics and probiotics, leaving aside studies on gene editing of probiotic microorganisms, although both options are sustainable alternatives to treat diseases. It is important to continue promoting the development of this biotechnological area for both types of microorganisms, as they represent a sustainable alternative within the industry in the coming years, mainly as a way to increase production while reducing mortality due to bacterial and viral pathogen attacks.

The use of genetically modified microorganisms (GMMs) in aquaculture represents a significant advancement in the field, offering new possibilities for improving fish health and production. However, the application of GMMs also raises important ethical, regulatory, and environmental considerations that must be addressed to ensure their safe and sustainable use.

## 5. Challenges and Future Perspectives

Aquaculture has experienced remarkable growth in recent decades, becoming a key component of global food production. However, this industry faces significant challenges that require innovative and sustainable solutions. Among these challenges are disease outbreaks, environmental impact, antibiotic resistance, and the need to enhance fish growth and nutrition while maintaining sustainability. The use of microorganisms, including probiotics, bacteriophages, and genetically modified microorganisms (GMMs), presents promising solutions, but their large-scale application still faces regulatory, technical, and societal hurdles.

### 5.1. Challenges in the Use of Microorganisms in Aquaculture

One of the primary challenges in aquaculture is the high incidence of infectious diseases caused by bacteria, viruses, and fungi, leading to economic losses estimated at over USD 6 billion annually [8]. While probiotics have shown effectiveness in improving fish health and survival, their impact is often strain-dependent and varies with environmental conditions [26]. Additionally, the survival and colonization capacity of probiotics in fish intestines remain unpredictable, necessitating further research to optimize formulations and delivery methods [9].

The indiscriminate use of antibiotics in aquaculture has led to the emergence of antimicrobial resistance (AMR), posing risks to both aquatic life and human health [34]. In response, bacteriophage therapy has been proposed as an alternative, showing promising results in targeting specific bacterial pathogens [205]. However, large-scale implementation of phage therapy requires overcoming challenges such as phage–host specificity, stability in aquaculture environments, and regulatory approval [206].

GMMs offer a new frontier in sustainable aquaculture, with applications ranging from improved disease resistance to enhanced nutrient absorption [5]. Despite their potential, public perception and regulatory constraints present barriers to commercialization [204]. Ethical concerns regarding the use of genetically modified organisms (GMOs) in food production remain a significant issue, requiring transparent risk assessments and public engagement strategies [11].

### 5.2. Future Perspectives

Advances in biotechnology, particularly in synthetic biology and gene editing, will play a crucial role in the future of aquaculture [12]. The development of CRISPR-based modifications in probiotics and bacteriophages could enhance their efficacy and specificity, reducing the need for antibiotics [207]. Additionally, novel encapsulation technologies may improve the stability and delivery of probiotics in fish feed [29].

Integrated multi-trophic aquaculture (IMTA) systems, which combine fish farming with the cultivation of complementary species (e.g., shellfish, algae), could provide a more sustainable approach to aquaculture by reducing environmental impact and improving nutrient recycling [24]. The incorporation of microbiome engineering strategies, including the selective enrichment of beneficial microbial communities, represents another promising avenue [30].

To fully realize the potential of microorganisms in sustainable aquaculture, interdisciplinary research efforts and collaboration between academia, industry, and regulatory bodies will be essential. Future policies should focus on balancing innovation with ecological and food safety considerations to ensure the responsible use of microbial technologies in aquaculture [7].

## 6. Conclusions

Probiotics microorganisms like bacteria, yeasts, and viruses are notable for their ability to positively alter the presence of beneficial microbiota and reduce pathogen microbiota, which often cause diseases with significant losses in fish aquaculture. These probiotics modulate various metabolic pathways and improve growth, survival, immune response, secretion of digestive enzymes, and production of macronutrients and micronutrients in fish muscle. However, to better understand the effects of these supplements, interdisciplinary research, including molecular analysis using various techniques (such as DNA and RNA sequencing and metabolite analysis), is suggested.

Global aquaculture is expanding due to rising demand and population growth, driving the need for higher-quality food. Probiotics, including bacteria, yeasts, and viruses, offer a sustainable solution to improve fish production. These probiotics can enhance the growth of globally important fish species and region-specific native species. Their application presents a promising strategy for increasing aquaculture efficiency.

Bacteriophages are a sustainable alternative to antibiotics for treating bacterial infections in aquaculture, with promising research results. Although methods like intraperitoneal injection show potential, they are not viable for large-scale use. Further research is needed to make bacteriophage treatments more practical and affordable for broader, long-term protection. In addition, the use of phage therapy for the treatment of bacterial co-infections needs to be evaluated, as such infections have been reported in wild and farmed fish.

Currently, it is not cost-effective to develop vaccines against all known pathogens that are identified, because vaccination in fish aquaculture focuses on pathogens that cause significant mortalities. In the future, vaccines produced to treat bacterial and viral pathogens in fish must be environmentally friendly and cost-effective for large-scale production. Furthermore, it is important that they are available to all types of fish producers, including small-scale producers.

The use of biotechnology through genetic modification of microorganisms, such as bacteria, yeasts, and viruses, has potential for disease management in fish aquaculture, which can effectively contribute to increased production by reducing mortality and providing organisms with better nutritional quality. However, genetic modification can often be rejected by the general public. It is important to implement measures on the importance of the use of genetically modified microorganisms (GMMs) as a sustainable alternative to increase aquaculture in the face of the growing food demand of the world’s population.

## Figures and Tables

**Figure 1 microorganisms-13-00485-f001:**
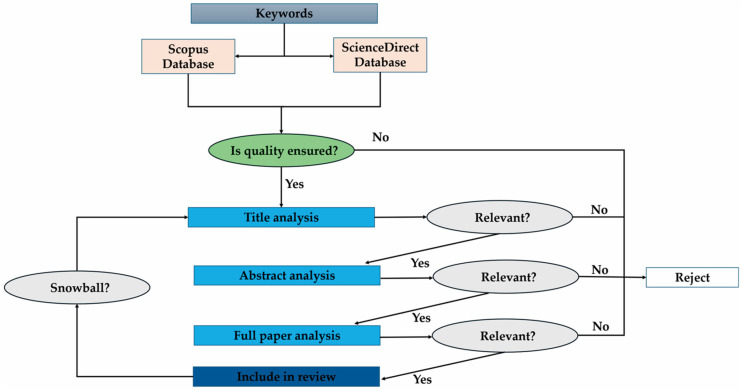
Flow chart for literature search using the PRISMA model.

**Figure 2 microorganisms-13-00485-f002:**
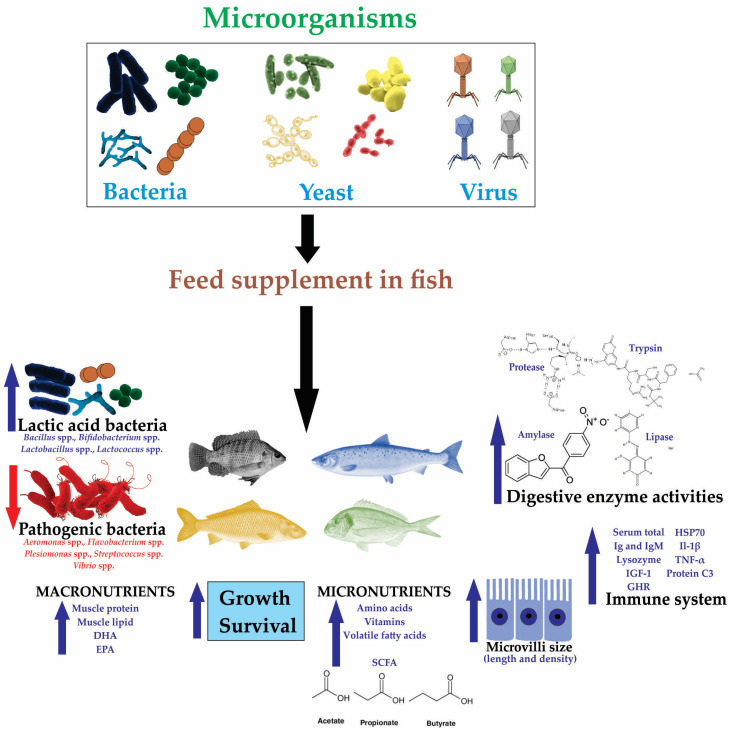
Effects of microorganisms—bacteria, yeast, and virus— applied in commercial fish. Blue arrows indicate increased effects. Red arrows indicate decreased effects. Black letters indicate the description of effects on fish.

**Figure 3 microorganisms-13-00485-f003:**
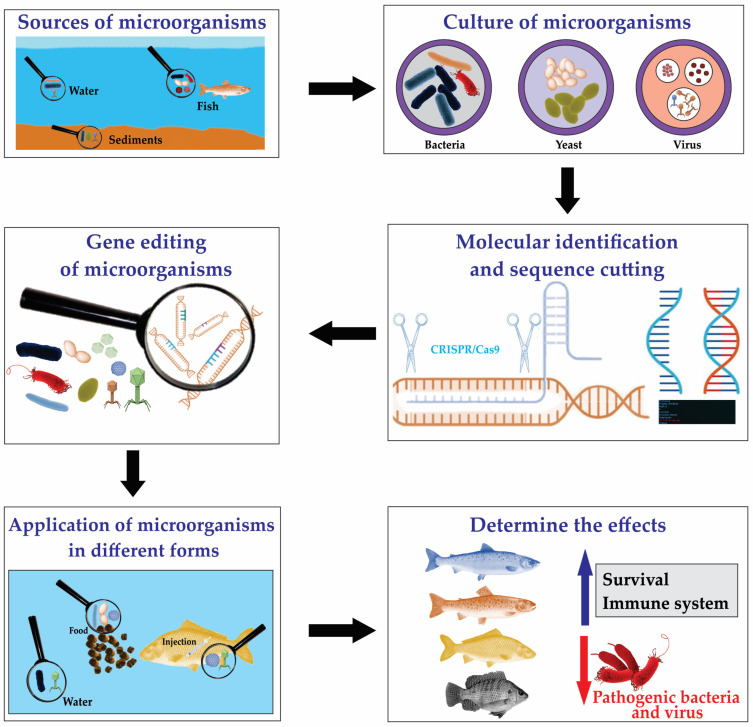
The process of searching, culturing, molecular identification, sequence cutting, gene editing, and applying genetically modified microorganisms (GMMs), including bacteria, yeast, and virus, in commercial fish to define their effects on survival and the immune system in the presence of pathogenic bacteria and viruses as an alternative to sustainable aquaculture. The blue arrow indicates increased effects and the red arrow indicates decreased effects.

**Table 1 microorganisms-13-00485-t001:** Keyword strings used in the literature search strategy and the corresponding number of matches.

Keyword Strings	Science Direct	Scopus	Used for	Documents Selected
EM + FA	12,971	946	Introduction	35
FA + P + SA	4647	1502	Introduction	14
B + FA	20,529	2211	Effects of bacteria on fish aquaculture	44
Y + FA	6971	234	Effects of yeasts on fish aquaculture	22
V + FA	10,756	581	Effects of bacteriophages on fish aquaculture	34
GMM + FA	8425	8	Use of genetically modified microorganisms in fish aquaculture	43

EM: effects of microorganisms; FA: fish aquaculture; P: probiotics; B: bacteria effects; Y: yeasts effects; V: bacteriophages effects; GMM: genetically modified microorganisms; SA: sustainable approach.

**Table 2 microorganisms-13-00485-t002:** Summary of research on the effects of bacteria probiotics supplements in fish with production and economic importance worldwide.

Fish Species	Microorganism	Concentration or Dose	Origin of Microorganism	Effect	Reference
Senegalese sole *Solea senegalensis*	*Shewanella* spp.	1 × 10^9^ CFU/g	Isolated from skin of *Sparus aurata*	↑ Growth	[68]
Grass carp *Ctenopharyngodon idella*	*Bacillus subtilis* Ch9	3 and 5 × 10^9^ CFU/kg	Isolated from intestine of *Ctenopharyngodon idella*	↑ Growth ↑ Bacteria *Bifidobacterium* and *Lactobacillus* ↑ Enzyme activity (protease, amylase and lipase)	[69]
Common carp *Cyprinus carpio*	*Bacillus coagulans*	1, 2 and 4 × 10^7^ CFU/g	Isolated from *C. carpio*	↑ Growth	[70]
Olive flounder *Paralichthys olivaceus*	*Lactobacillus plantarum* FGL0001	1 × 10^7^ CFU/g	Isolated from hindgut of *P. olivaceus*	↑ Growth	[71]
Siberian sturgeon *Acipenser baerii*	*Lactobacillus plantarum*	1 × 10^8^ CFU/g	Isolated from the digestive tracts of *Oncorhynchus* *mykiss*	↑ Growth and innate immune response	[72]
Nile tilapia *Oreochromis niloticus*	*Lactobacillus plantarum* AH 78	1%	Isolated from corals along the Egyptian coasts of the Mediterranean Sea	↑ Growth, immune response and survival ↑ Total protein in muscle	[73]
Goldfish *Carassius auratus*	*Lactobacillus helveticus*	3 × 10^7^ CFU/g	Isolated from Indian traditional fermented food	↑ Growth	[74]
Nile tilapia *Oreochromis niloticus*	*Lactobacillus plantarum* L-137	50 ppm	Isolated from a fermented fish and rice dish	↑ Growth	[75]
Common carp *Cyprinus carpio*	*Lactobacillus delbrueckii*	1 × 10^6^ CFU/g	From Angel Company, Wuhan, China	↑ Growth	[76]
Nile tilapia *Oreochromis niloticus*	*Bacillus* spp.	1 × 10^9^ CFU/g	From Pond Care, SKF BioJoc Fish Probiotic, Bangladesh	↑ Growth	[77]
Striped catfish *Pangasius hypophthalmus*	*Bacillus subtilis*	1 × 10^8^ and 1 × 10^10^ CFU/g	From ECOSH, Estonian	↑ Growth, body protein and digestive enzymes (amylase and protease)	[78]
Tropical gar *Atractosteus tropicus*	*Lactococcus lactis* PH3-05	1 × 10^4^, 1 × 10^6^ and 1 × 10^8^ CFU/g	Isolated from the intestine of an adult male of *A. tropicus*	↑ Growth, survival and digestive enzymes	[79]
Olive flounder *Paralichthys olivaceus*	*Lactococcus lactis*	1 × 10^8^ CFU/mL	Isolated from the intestine of *P. olivaceus*	↑ Growth performance parameters	[80]
Catfish *Clarias gariepinus*	*Bacillus subtilis* and *Lactobacillus casei*	1 × 10^8^ CFU/mL	Not specified	↑ Growth and immune system	[81]
Asian catfish *Pangasius bocourti*	*Bacillus aerius* B81e and *Lactiplantibacillus paraplantarum* L34b-2	1 × 10^7^ CFU/g	Strain B81e isolated from the intestine of *P. bocourti* Strain L34b-2 isolated from fermented food samples	↑ Growth performance parameters and immune system (lysozymes)	[82]
Cobia *Rachycentron canadum*	*Pantoea agglomerans* RCS2	1 × 10^10^ and 1 × 10^12^ CFU/mL	Isolates from *R. canadum*	↑ Growth and activity of digestive enzymes	[83]
Common carp *Cyprinus carpio*	*Streptomyces chartreusis*	1 × 10^6^ and 1 × 10^7^ CFU/g	Isolated from soil ecosystem	↑ Growth performance parameters ↑ Serum total Ig and lysozyme activity	[84]
Nile tilapia *Oreochromis niloticus*	* Bacillus coagulans * DSM 32016	0.02, 0.04, and 0.08%	Isolated from canned tomatoes	↑ Immune-related genes, including liver IGF-1, GHR, HSP70, IL-1β, and TNF-α and IL-1β and intestinal C-lysozyme and TNF-α	[85]
Common carp *Cyprinus carpio*	*Lactobacillus acidophilus* ATCC 4356	1 × 10^6^ CFU/kg	Isolated from human intestine	↑ Growth performance parameters	[86]
Common carp *Cyprinus carpio*	*Enterococcus casseliflavus*	1 × 10^12^ CFU/kg	Isolated from the intestine of *C. carpio*	↑ Growth performance parameters	[87]
Rainbow trout *Oncorhynchus mykiss*	*Lactobacillus rhamnosus* ATCC 7469	1 × 10^9^ CFU/kg	Purchased from Persian Type Culture Collection	↑ Growth performance parameters	[88]
Nile tilapia *Oreochromis niloticus*	Mixture of *Bacillus, Bifidobacterium, Enterococcus, Lactobacillus, Pediococcus* sp. and *B. subtilis*	7 × 10^10^ CFU/kg	Provided by Biomart Nutrição Animal Importação e Exportação LTDA	↑ Growth performance parameters	[89]
African catfish *Clarias gariepinu*	Probiotic sporothermine (*Bacillus subtilis* and *B. licheniformis*)	0.2%	Provided by Ulyanovsk State University, Russia	↑ Vitamins B3, B5, B6, C, and E in the muscle	[90]
Amberjack *Seriola dumerili*	*Lactococcus lactis* K-C2	2 × 10^10^ CFU/g	Isolated from fermented vegetables	↑ Amino acids in the gut content	[61]
Javanese carp *Puntius gonionotus*	*Enterococcus faecalis*	2 × 10^7^ CFU/g	Isolated from the intestine of *Channa striatus*	↑ SCFA (propionic and butyric acids)	[91]
Caspian Roach *Rutilus frisii* *kutum*	*Pediococcus* *acidilactici and Lactococcus lactis*	1 × 10^7^ and 1 × 10^10^ CFU/g	*P. acidilactici* comercial Bactocell^®^ (Lallemand animal nutrition, Blagnac, France). *L. lactis* isolated from juvenile sturgeon *Acipenser persicus* gut	↑ SCFA (acetic and butyric acids)	[92]
Pacific mackerel *Pneurnatophorus japonicus*	*Shewanella putrefaciens*	2 × 10^10^ viable cells/mL	Isolated from the intestine of contents of *P. japonicus*	↑ EPA	[93]
Siberian sturgeon *Acipenser baerii*	*Lactobacillus plantarum,* *L. delbrueckii* subsp. *Bulgaricus,* *L. acidophilus,* *L. rhamnosus,* *Bifidobacterium bifidum* *Streptococcus salivarius* subsp. *Thermophilus and* *Enterococcus faecium*	0.01%	Commercial probiotic name of Protexin^®^ (ADM Protexin Limited, Somerset, UK)	↑ DHA and EPA	[94]
Rainbow trout *Oncorhynchus mykiss*	*Enterobacter* sp. (strain C6-6)	1.03 × 10^7^ CFU/g	Isolated from the intestine of *O. mykiss*	↑ Entericidin protein ↑ Protection against *Flavobacterium psychrophilum*	[95]
Nile tilapia *Oreochromis niloticus*	*Bacillus* *amyloliquefaciens*	3.0 × 10^3^ and 7.9 × 10^4^CFU/g	Commercial probiotic mix (Enviva^®^ PRO 202 GT, Danisco Animal Nutrition, Wiltshire, UK)	↑ Volatile fatty acids	[96]
Nile tilapia *Oreochromis niloticus*	*Lactococcus lactis* subsp. *lactis JCM5805*	1 × 10^8^ CFU/g	Provided by China General Microbiological Culture Collection Center (CGMCC)	↑ Volatile fatty acids	[97]

The symbol (↑) represents increased parameters of growth, survival, immune response (serum total Ig, lysozyme, IGF-1, GHR, HSP70, IL-1β, TNF-α), secretion of digestive enzymes (protease, amylase and lipase), production of macronutrients (entericidin protein, muscle protein, DHA, EPA) and micronutrients (amino acids, SCFA, vitamins, volatile fatty acids) in fish muscle, lactic acid bacteria, and protection against pathogenic bacteria relative to the control treatment at the time of the study. Abbreviations: Ig: immunoglobulin, IGF-1: insulin-like growth factor 1, GHR: Growth hormone receptor, HSP70: heat shock protein (Hsp)-70, IL-1β: Interleukin-1 beta, TNF-α: Tumor necrosis factor, DHA: Docosahexaenoic acid, EPA: Eicosapentaenoic acid, SCFA: Short-Chain Fatty Acid, CFU: Colony Forming Units.

**Table 3 microorganisms-13-00485-t003:** Summary of research on the effects of yeasts probiotics supplements in fish with production and economic importance worldwide.

Fish Species	Microorganism	Concentration or Dose	Origin of Microorganism	Effect	Reference
Gibel carp *Carassius auratus gibelio*	*Saccharomyces cerevisiae*	4 and 6%	Purchased from Enhalor Biotechnology Company (Beijing, China)	↑ Immune system (IL-1β) and survival rate in presence of pathogen *Aeromonas hydrophila*	[105]
Common carp *Cyprinus carpio*	*Saccharomyces cerevisiae*	1.5%	Obtained from the local markets of Basrah, Turkey	↑ Growth performance parameters (total weight gain, relative growth rate and feed conversion efficiency)	[106]
Largemouth bass *Micropterus salmoides*	*Saccharomyces cerevisiae*	3%	Culture (from Beijing Enhalor International Tech Co., Ltd., Beijing, China)	↑ Growth ↑ Abundance of beneficial bacteria (*Lactobacillus*, *Bacillus* and *Bifidobacterium*) ↓ Abundance of potential pathogenic bacteria *Plesiomonas*	[107]
Nile tilapia *Oreochromis niloticus*	*Saccharomyces cerevisiae*	0.5%	Obtained as a commercial preparation (Perfect^®^, Dejo Co., Ltd., Bangkok, Thailand)	↑ Growth performance	[108]
Nile tilapia *Oreochromis niloticus*	*Saccharomyces cerevisiae*	0.4%	Culture (from Angel Yeast Co., Ltd., Yichang, China)	↑ Growth ↑ Length, width and area of villus in gut	[109]
Nile tilapia *Oreochromis niloticus*	*Saccharomyces cerevisiae*	0.1, 0.2, and 0.3%	Hilyses^®^ commercial products (ICC Industrial Comércio Exportaçãoe Importação SA, São Paulo, Brazil)	↑ Growth and weight gain	[110]
Nile tilapia *Oreochromis niloticus*	*Saccharomyces cerevisiae*	2, 2.5 and 3%	From palm wine	↑ Growth ↑ Resistance to pathogen *Aeromonas hydrophila*	[103]
Nile tilapia *Oreochromis niloticus*	*Saccharomyces boulardii and Bifidobacterium bifidum*	*Saccharomyces boulardii* (1 × 10^10^ CFU/g), *Bifidobacterium bifidum* (1.5 × 10^8^ CFU/mL) and mixture of both	Acquired from the Iranian Biological Resource Center (Tehran, Iran)	↑ Growth and immune responses	[111]
Leopard grouper *Mycteroperca rosacea*	*Debaryomyces hansenii*	1 × 10^6^ CFU/g	Isolated from the intestine of *O. mykiss*	↑ Immune system and resistance against pathogen *Amyloodinium ocellatum*	[112]
Gilthead seabream *Sparus aurata*	*Debaryomyces hansenii*	1.1%	Isolated from the intestine of *O. mykiss*	↑ Growth ↓ Abundance of opportunistic bacteria *Pseudomonas* spp. and *Acinetobacter* spp.	[102]
Sea bass *Dicentrarchus labrax*	*Debaryomyces hansenii*	1.1%	Isolated from the intestine of *O. mykiss*	↑ Survival and digestive enzymes (trypsin and lipase)	[113]
Pacific red snapper *Lutjanus peru*	*Yarrowia lipolytica*	1 × 10^8^ CFU/mL	Isolated from the world’s largest open-air saltern known in Baja California Sur, Mexico	↑ Innate immune and antioxidant enzyme activities in presence of pathogen *Vibrio parahaemolyticus*	[114]
Nile tilapia *Oreochromis niloticus*	*Yarrowia lipolytica*	3, 5 and 7%	Provided from Federal University of Rio Grande do Sul (UFRGS, Porto Alegre, Brazil)	↑ Digestive enzymes (chymotrypsin, trypsin and sucrose) ↑ Protein and lipid contents in fish muscle	[115]
Rainbow trout *Oncorhynchus mykiss*	*Yarrowia lipolytica*	2 and 5%	Isolated from sewage from a wastewater treatment plant in Uppsala, Sweden	↑ Expression of immune genes	[104]
Nile tilapia *Oreochromis niloticus*	*Yarrowia lipolytica*	3, 5 and 7%	Provided from Federal University of Rio Grande do Sul (UFRGS, Porto Alegre, Brazil)	↑ Growth promoter and immunostimulant	[116]
Golden Pompano *Trachinotus ovatus*	*Rhodotorula mucilaginosa*	1, 2, 3, 4, 5 and 8%	Provided by Xinhailisheng technology company (Guangzhou, China)	↑ Growth, lysozyme activity and resistance 100% survival rate against the pathogen *Vibrio harveyi*	[117]
Nile tilapia *Oreochromis niloticus*	*Rhodotorula mucilaginosa*	1%	Supplied by South China Sea Fisheries Research Institute, Chinese Academy of Fishery Sciences	↑ Growth and protein content in the whole-body ↑ Immune system (lysozyme) and villi height of mid-intestine ↑ Survival rate in presence of pathogen *Streptococcus iniae*	[101]
Nile tilapia *Oreochromis niloticus*	*Sporidiobolus pararoseus*	1 and 2%	By product of the biodiesel production process	↑ Growth ↑ Immune response against pathogen *Streptococcus agalactiae*	[118]
Gilthead seabream *Sparus aurata*	*Sterigmatomyces halophilus*	0.55 and 1.1%	Isolated from the world’s largest open-air saltern in Baja California Sur, Mexico	↑ Trypsin and immune related gene expression (IL-1β, TNF-α, IgM, C3 and lysozyme), in presence of pathogen *Vibrio parahaemolyticus*	[119]

The symbols represent increment (↑) or decrement (↓) on parameters of growth, survival, immune response (lysozyme, IL-1β, TNF-α, IgM, protein C3), secretion of digestive enzymes (chymotrypsin, trypsin, and sucrose), production of macronutrients (muscle protein, DHA, EPA) and micronutrients (amino acids, SCFA, vitamins, volatile fatty acids) in fish muscle, lactic acid bacteria, and protection against pathogenic bacteria relative to the control treatment at the time of the study. Abbreviations: IGF-1: insulin-like growth factor 1, GHR: growth hormone receptor, HSP70: heat shock protein (Hsp)-70, IL-1β: Interleukin-1 beta, TNF-α: tumor necrosis factor, IgM: immunoglobulin M, CFU: colony forming units.

**Table 4 microorganisms-13-00485-t004:** Summary of research on the effects of phage therapy for the control of infections with the main bacterial pathogens of fish.

Fish Species	Phage Strain Name/Virus Taxonomic Family	Concentration or Dose	Origin of Microorganism	Effect	Reference
Gram-negative bacteria *Aeromonas hydrophila*					
Rohu *Labeo rohita*	AhFM11/*Straboviridae*	I: 1.5 × 10^5^ PFU/fish B: 1.5 × 10^7^ PFU/mL F: 1.5 × 10^7^ PFU/g of feed pellets. MOI = 1000	From river	I: showed 100% survival, B: 95% survival, F: 93% feeding of phage top-coated feed	[136]
*Aeromonas salmonicida*					
Senegalese sole *Solea senegalensis*	AS-A/*Myoviridae*	W: 1 × 10^10^ PFU/mL (MOI 100)	From sewage	No mortality, in the control group mortality 36%	[141]
*Cytrobacter freundii*					
Common carp *Cyprinus carpio*	IME-JL8/*Siphoviridae*	I: 1 × 10^8^ PFU/mL	From sewage	Decrement pro-inflammatory cytokines	[142]
*Edwardsiella piscicida*					
Zebrafish *Danio rerio*	EPP-1/*Heunggongvirae*	I: MOI of 0.1, 1, 5, and 10)	From aquaculture wastewater	Treatment with MOI 1 significantly improved survival, similar in effectiveness to the florfenicol therapy group	[143]
*Edwardsiella tarda*					
Zebrafish *Danio rerio*	ETP-1/*Podoviridae*	B: 9.85 × 10^8^ PFU/mL	From fish farm water	The survival rate was higher in phage-exposed fish (68%) compared to that of the control (18%) until 4 days post-challenge	[144]
*Pseudomonas plecoglossicida*					
Ayu fish *Plecoglossus altivelis*	PTH-9802/*Myoviridae* PPpW-3 and PPpW/*Podoviridae*	F: 1 × 10^7^ PFU/g feed	From farm water	Survival rate of 78%	[145]
*Vibrio harveyi*					
Turbot *Scophthalmus maximus*	PVHp5/*Au-tographiviridae* PVHp8/*Myoviridae*	F: phage cocktail (MOI 1,10, 100)	From water	80% survival at MOI 10–100, normal fish growth	[146]
*Flavobacterium psychrophilum*					
Rainbow trout *Oncorhynchus mykiss*	FpV4/*Podoviridae* FPSV-D22/*Siphoviridae*	F: bacteriophage cocktails by spraying (1.6 × 10^8^ PFU/g) or by irreversible immobilization (8.3 × 10^7^ PFU/g). I: 1.7 × 10^7^ PFU/fish). W: 1 × 10^5^−1 × 10^8^ PFU/mL (MOI = 1).	From fecal water samples From rainbow trout organs	I: 80% survival compared to the control group of 57%	[139]
*Plesiomonas shigelloides*					
Grass carp *Ctenopharyngodon idella*	PSP01/*Siphoviridae*	I	From intestine of *C. idella*	Strong protective effect, increased survival by 33%	[147]
Gram-positive bacteria *Streptococcus agalactiae*					
Nile tilapia *Oreochromis niloticus*	1A/*Myoviri-dae*	I: 100 μL of phage) F: 3 mL/10 g feed B: 200 μL phage/L water (MOI 1)	From fish farm water	B: highest protection with 80% survival compared to applications I and F with 70% and 50% animal survival	[125]
*Lactococcus garvieae*					
Yellowtail *Seriola quinqueradiata*	PLG-Y16/*Siphoviridae*	I: 1 × 10^7^.5 PFU/fish F: 2% fish body feeding rate	From municipal wastewater	Both administrations with potential for use of phage therapy to control the disease	[148]

Abbreviations: PFU: plaque forming units, MOI: multiplicity of infection I = intraperitoneal injection, W = addition in the culture water. B = immersion bath, F = addition in food.

## Data Availability

No new data were created or analyzed in this study. Data sharing is not applicable to this article.

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
