# Peer review of "Effects of Microorganisms in Fish Aquaculture from a Sustainable Approach: A Review"

_microorganisms, 2025, doi:10.3390/microorganisms13030485_

Round 1

Reviewer 1 Report

Comments and Suggestions for Authors

The manuscript entitled "Effect of microorganisms in fish aquaculture from a sustainable approach: a review" submitted for consideration in the journal Microorganisms deals with a very interesting topic in aquaculture, such as the use of various microorganisms (probiotics) in fish farming as a sustainable alternative to traditional treatments for infectious diseases.

However, I consider that the manuscript, in its current form, does not provide new information on the subject, considering the high number of articles already published. For example, in a search in the main bibliographic databases, at least seven review articles appear on the subject of phage terapy in aquaculture in the last five years, of which only one is cited in the reference list of the article.

In addition, the manuscript lacks a section on materials and methods in which it is explained how the information was searched and how the articles were or not considered for the review.

Most of the information provided in section 3: "Use of genetically modified microorganisms in fish aquaculture" refers to the modification of pathogenic microorganisms in aquaculture for the production of vaccines and in my point of view would be outside the scope of the article.

For all these reasons, I consider that the manuscript is not suitable for publication in the journal Microorganisms.

Author Response

Dear Reviewer,

First, I would like to take this opportunity to send you a cordial greeting and, at the same time, thank you for taking part of your time to review the manuscript.

Regarding your comments/concerns identified as possible opportunities for improvements to the document, I will respond below with the changes made, which are indicated by yellow shading in the new version of the document.

The manuscript entitled "Effect of microorganisms in fish aquaculture from a sustainable approach: a review" submitted for consideration in the journal Microorganisms deals with a very interesting topic in aquaculture, such as the use of various microorganisms (probiotics) in fish farming as a sustainable alternative to traditional treatments for infectious diseases.

However, I consider that the manuscript, in its current form, does not provide new information on the subject, considering the high number of articles already published. For example, in a search in the main bibliographic databases, at least seven review articles appear on the subject of phage therapy in aquaculture in the last five years, of which only one is cited in the reference list of the article.

Response. Thank you very much for your comment. However, we differ in your idea because, although there are many different articles published on this subject, in this article we present the potential of different microorganisms such as bacteria, yeasts and viruses as a sustainable alternative to fish aquaculture compared to the use of unsustainable practices such as the use of antibiotics. We present this information together, and not separately (as they do in most works of this type). In addition, we mention and analyzed the potential alternative in the development of the use of genetically modified microorganisms in this productive activity. Also, when consciously analyzing the sections of this manuscript, we considered adding a subtheme entitled Challenges and future perspectives, previously to the conclusion section.

In addition, the manuscript lacks a section on materials and methods in which it is explained how the information was searched and how the articles were or not considered for the review.

Response. Thank you for your comment. We agree that it is necessary to add in the manuscript the strategy used for the search of the information. We added this strategy in a section entitled Methodology Applied for the Literature Review, which is after the Introduction section of the manuscript. We do not consider it prudent to add a section entitled Materials and Methods in a review article, as we believe that a section with this name applies only to articles about original research manuscripts and Case reports, for example.

Most of the information provided in section 3: "Use of genetically modified microorganisms in fish aquaculture" refers to the modification of pathogenic microorganisms in aquaculture for the production of vaccines and in my point of view would be outside the scope of the article.

Response. Thank you for your comment. I agree that most of the information in this section is on the modification of pathogenic microorganisms. The reason for this is because most of the work in this area is focused on dealing directly with pathogenic organisms, which are the ones that cause the main production and monetary problems, and it is often difficult to manage them with common antibiotics and probiotics. So, they leave aside experimenting with editing such probiotics and choose in these papers to analyze the alternative of modifying pathogenic microorganisms to treat diseases mainly by means of vaccines. However, although the number of studies differs, we consider keeping this topic within the manuscript, as the use of both probiotic and pathogenic GMMs represent a sustainable alternative in the medium term, if research of this type continues to grow, and taking into account, of course, the advances in terms of legislation on their use.

In addition, thanks to your timely comment, we considered improving the wording of the section on the increased focus on the use of genetically modified pathogens for fish aquaculture. This was added to the manuscript at the end of the section “Use of genetically modified microorganisms in fish aquaculture”.

For all these reasons, I consider that the manuscript is not suitable for publication in the journal Microorganisms.

Response. Thank you for your comment. An apology. However, we differ in your idea because we consider that the structure of the different sections developed and analyzed in this manuscript are adequately adjusted to the Special Issue of the journal Microorganisms entitled Aquatic Microorganisms and Their Application in Aquaculture.

In summary, some of the most notable changes we made to the manuscript were: changed the abstract, added a subtheme Methodology Applied for the Literature Review explaining how the information was sought and how papers were considered for addition to the review, and added a section on Challenges and future perspectives.

The authors greatly appreciate the comments of the reviewer, since have improved the quality of the presentation of the work. Thank you for your time.

Best regards

Reviewer 2 Report

Comments and Suggestions for Authors

1. There have been many articles on the review of beneficial microorganisms in aquatic animals, so what is unique about this manuscript? Where are the highlights? These will be better introduced in the introduction.

2. L56-61: The parasitic pathogens of aquatic animals should also be reviewed.

3. L179: “2. Materials and Methods”. It is not appropriate to use such a title in the review article.

4. L155-159: Genera's Latin name needs to be italicized. You should check the manuscript and correct such questions.

5. L167-169: The functions of these beneficial bacteria also include improving intestinal health and regulating intestinal microbial community.

6. It is better to review the methods of obtaining beneficial microorganisms.

7. The use of Latin scientific names of species needs to be standardized, and you should pay attention to the abbreviations of generic names.

8. Table 3: The virus name should be clearly written, not the code.

9. Conclusions: The content of the conclusion should be reduced, and the most important research progress and future prospect should be summarized.

Author Response

Dear Reviewer,

First, I would like to take this opportunity to send you a cordial greeting and, at the same time, thank you for taking part of your time to review the manuscript.

Regarding your comments/concerns identified as possible opportunities for improvements to the document, I will respond below with the changes made, which are indicated by yellow shading in the new version of the document.

  1. There have been many articles on the review of beneficial microorganisms in aquatic animals, so what is unique about this manuscript? Where are the highlights? These will be better introduced in the introduction.

Response. Thank you very much for your comment. You are right about the literature on this type of work. However, although there are many different articles published on this subject, in the present article we present the potential of different microorganisms such as bacteria, yeasts and viruses as a sustainable alternative to the activity of fish aquaculture compared to the use of unsustainable practices such as the use of antibiotics and as a necessity for an activity in continuous growth due to the demand for resources by the growing world population. We also mention and analyze the potential alternative in the development and application of genetically modified microorganisms in this productive activity. Furthermore, in the sustainable alternatives proposed, it is emphasized that these types of practices correspond to modern trends for the development and growth of aquaculture on a global scale. These points raised about the relevance of the present research work were added to in the introduction.

  1. L56-61: The parasitic pathogens of aquatic animals should also be reviewed.

Response. Thank you very much for this comment. We have added information in the manuscript on microscopic parasitic pathogens that also cause significant havoc in aquaculture.

  1. L179: "2. Materials and Methods". It is not appropriate to use such a title in the review article.

Response. Thank you for your comment. I agree with the title error. There was an error in placing the proper name in this section when adjusting to the journal format, as item 2 is actually titled " Effects of microorganisms in fish aquaculture ". The change has already been made in the manuscript. In addition, a subtheme entitled Methodology Applied for the Literature Review has been added to section 2. Therefore, "Effects of microorganisms in fish aquaculture" is now section 3 of the manuscript.

  1. L155-159: Genera's Latin name needs to be italicized. You should check the manuscript and correct such questions.

Response. Thank you very much for this comment. I agree with your suggestion. These issues have been corrected throughout the manuscript.

  1. L167-169: The functions of these beneficial bacteria also include improving intestinal health and regulating intestinal microbial community.

Response. We agree with your comment. Such suggested functions were added in the drafting of the manuscript.

  1. It is better to review the methods of obtaining beneficial microorganisms.

Response. Thank you for your comment. This part was reviewed again in each article analyzed and the necessary adjustments were made taking into account the layout of this information present in such works.

  1. The use of Latin scientific names of species needs to be standardized, and you should pay attention to the abbreviations of generic names.

Response. Thank you very much for your comment. These suggestions were analyzed and corrected throughout the manuscript.

  1. Table 3: The virus name should be clearly written, not the code.

Response. Thank you for pointing this out. We agree with this comment. We consult each article again to place the requested information. Therefore, in this Table, new information on the bacteriophages were added to the taxonomic family level because virus names generally are at family level. This information was added in the column called Phage strain name/Virus taxonomic family, which was used as a substitution for the Virus column in this Table. The change has already been made in the manuscript. In addition, as when the subtopic entitled Methodology Applied for the Literature Review was added in section 2, we placed a table. Therefore, the original Table 3 is now Table 4 in the manuscript.

  1. Conclusions: The content of the conclusion should be reduced, and the most important research progress and future prospect should be summarized.

Response. Thank you very much for your comments on this part of the manuscript. In consciously analyzing this section, we considered adding a subtheme entitled Challenges and Future perspectives, prior to the conclusion section. In addition, we reduced the content of the conclusion.

In summary, in addition to addressing your suggested changes in the drafting of this manuscript, other changes were made to the manuscript such as: Change of abstract, addition of the subtheme Methodology Applied for the Literature Review explaining how the information was sought and how papers were considered for addition to the review, and addition of a section on Challenges and Future Perspectives.

The authors greatly appreciate the comments of the reviewer, since have improved the quality of the presentation of the work. Thank you for your time and effort.

Best regards

Reviewer 3 Report

Comments and Suggestions for Authors

The review by Jesús Mateo Amillano-Cisneros et al. covers the role of microbial communities in aquaculture, which is the fastest-growing food production sector.

The authors have analyzed recent literature data, including FAO statistics and studies from 2022-2024, ensuring the review reflects the latest trends and scientific advancements in aquaculture microbiology.

I also would like to acknowledge that the authors placed an emphasis on sustainability-based approaches in aquaculture, which corresponds to modern trends in agricultural development.

In my opinion, the review will be frequently cited and its publication will make a significant contribution to Microorganisms journal.

However, I need to acknowledge that some essential corrections need to be made prior to the final decision on the acceptance of the paper.

1. L. 129. Please consider writing another heading title. There should not be an M&M section in Review papers.

2. Figures are uploaded to the section entitled "Original Images for Blots/Gels" in the submission system, however, there are no blots or gels presented in the figures. Please check this accordingly.

3. The most important issue I would like to highlight is the lack of critical analysis of discussed studies. At the moment, the review summarizes the findings of existing studies. I strongly recommend adding a critical evaluation of the limitations of the discussed studies and how to overcome them in the future, which will benefit the manuscript.

Author Response

Dear Reviewer,

First, I would like to take this opportunity to send you a cordial greeting and, at the same time, thank you for taking part of your time to review the manuscript.

Regarding your comments/concerns identified as possible opportunities for improvements to the document, I will respond below with the changes made, which are indicated by yellow shading in the new version of the document.

The review by Jesús Mateo Amillano-Cisneros et al. covers the role of microbial communities in aquaculture, which is the fastest-growing food production sector.

The authors have analysed recent literature data, including FAO statistics and studies from 2022- 2024, ensuring the review reflects the latest trends and scientific advancements in aquaculture microbiology.

I also would like to acknowledge that the authors placed an emphasis on sustainability-based approaches in aquaculture, which corresponds to modern trends in agricultural development.

In my opinion, the review will be frequently cited and its publication will make a significant contribution to Microorganisms journal.

However, I need to acknowledge that some essential corrections need to be made prior to the final decision on the acceptance of the paper.

Response. Thank you very much for your comments on the content of the article

  1. L. 129. Please consider writing another heading title. There should not be an M&M section in Review papers.

Response. Thank you very much for your comment. We agree on the title error. There was an error in placing the proper name in this section when adjusting to the journal format, as item 2 is actually titled "Effects of microorganisms in fish aquaculture". The change has been made to the manuscript. In addition, a subtheme entitled Methodology Applied for the Literature Review has been added to section 2. Therefore, "Effects of microorganisms in fish aquaculture" is now section 3 of the manuscript.

  1. Figures are uploaded to the section entitled "Original Images for Blots/Gels" in the submission system, however, there are no blots or gels presented in the figures. Please check this accordingly.

Response. Thank you very much for your comment. An apology for this error. However, when submitting the various files that were indicated to us on the platform in the submission system, we could not find the section "Original Images for Blots/Gels". We only noticed the section Figures, Graphics, Images, section in which we were only allowed to upload one file at a time. Therefore, the two figures placed in the manuscript I merged them into one .pdf file and added them in this way. I would very much appreciate some support from you or the committee of editors in this part of uploading the figures, if necessary.

  1. The most important issue I would like to highlight is the lack of critical analysis of discussed studies. At the moment, the review summarizes the findings of existing studies. I strongly recommend adding a critical evaluation of the limitations of the discussed studies and how to overcome them in the future, which will benefit the manuscript.

Response. Thank you very much for your comments. By looking more consciously at the different sections of the manuscript and the studies examined in each of them, we discussed more critically the information contained in these papers and highlighted the limitations of this research in the different sections worked on in the manuscript. For example, in the introduction section we added another important challenge for aquaculture production is the presence of co-infections with different pathogenic microorganisms, which can further increase disease demands.

In addition, based on these changes made and discussed throughout the manuscript, we considered adding a sub-theme entitled Challenges and Future Perspectives, prior to the conclusion section. Also, in the manuscript we added a subtheme mentioning the methodology for processing the information from the articles used for this review. This subtheme is entitled "Methodology Applied for the Literature Review" and is located after the Introduction section.

In summary, in addition to addressing your suggested changes in the drafting of this manuscript, other changes were made to the manuscript such as: Change of abstract, addition of the subtheme Methodology Applied for the Literature Review explaining how the information was sought and how papers were considered for addition to the review, and addition of a section on Challenges and Future Perspectives.

The authors greatly appreciate the comments of the reviewer, since have improved the quality of the presentation of the work. Thank you for your time and effort.

Best regards

Round 2

Reviewer 1 Report

Comments and Suggestions for Authors

In this new revised version of the article entitled: "Effects of microorganisms in fish aquaculture from a sustainable approach: A Review"; the authors have included most of the suggested corrections, so the reviewer think that the article can be considered for publication. However, I suggest the following minor corrections:

The authors have included a section on the methodology used in the literature review (section 2), but the reviewer think that the inclusion and exclusion criteria should be better explained, as well as reflecting in Table 1 the total number of articles finally included and excluded in each combination of keywords.

Section 3.2 (line 318 and following) refers to the effects of yeasts in fish aquaculture and begins with the definition of yeast as: "unicellular eukaryotic microorganisms"; therefore any reference in this section, both in the text and in Table 3, to the genus Aspergillus should be deleted as this is a multicellular filamentous fungus.

In reference 138 (lines 1076-1077) the year of publication has been omitted.

Author Response

In this new revised version of the article entitled: "Effects of microorganisms in fish aquaculture from a sustainable approach: A Review"; the authors have included most of the suggested corrections, so the reviewer think that the article can be considered for publication. However, I suggest the following minor corrections:

Response. Thank you very much for your comments on the content of the article.

We will respond below with the news changes made, which are indicated by green shading in the new version of the document.

The authors have included a section on the methodology used in the literature review (section 2), but the reviewer think that the inclusion and exclusion criteria should be better explained, as well as reflecting in Table 1 the total number of articles finally included and excluded in each combination of keywords.

Response. Thank you very much for your comment. The inclusion and exclusion criteria previously cited in figure 1 have been better explained. The changes are in the manuscript indicated by green shading.

Regarding Table 1, the adjustment was made regarding the documents included in each keyword combination. Therefore, we added a new column in table 1 entitled “Documents selected”. However, we did not consider adding the list of excluded articles to the table, as this would only subtract the number of documents found in the cited databases from the number of articles included in the new column, which would be adding two more columns to the table, and we believe would be cluttered and reduce the impact of the title of the table which is the keyword search and the documents finally used.

Section 3.2 (line 318 and following) refers to the effects of yeasts in fish aquaculture and begins with the definition of yeast as: "unicellular eukaryotic microorganisms"; therefore any reference in this section, both in the text and in Table 3, to the genus Aspergillus should be deleted as this is a multicellular filamentous fungus.

Response. We are very grateful for this comment. Any reference to the genus Aspergillus has been deleted.

In reference 138 (lines 1076-1077) the year of publication has been omitted.

Response. Thank you for your comment. The year of publication has been added.

The authors greatly appreciate the comments of the reviewer. Thank you for your time and effort.

Best regards

Reviewer 2 Report

Comments and Suggestions for Authors

I have no other concerns.

Comments on the Quality of English Language

The English could be improved to more clearly express the research.

Author Response

I have no other concerns.

Response. Thank you very much for your comments on the content of the article.

The English could be improved to more clearly express the research.

Response. We agree. Some wording was redrafted in the manuscript to improve the wording.

The authors greatly appreciate the comments of the reviewer. Thank you for your time and effort.

Best regards

Reviewer 3 Report

Comments and Suggestions for Authors

The authors addressed all the comments.

Author Response

The authors addressed all the comments.

Response. The authors greatly appreciate the comments of the reviewer. Thank you for your time and effort.

Best regards